# Shannon Entropy and Diffusion Coefficient in Parity-Time Symmetric Quantum Walks

**DOI:** 10.3390/e23091145

**Published:** 2021-08-31

**Authors:** Zhiyu Tian, Yang Liu, Le Luo

**Affiliations:** 1School of Physics and Astronomy, Sun Yat-Sen University, Zhuhai 519082, China; tianzhy6@mail2.sysu.edu.cn (Z.T.); liuyang59@mail.sysu.edu.cn (Y.L.); 2Center of Quantum Information Technology, Shenzhen Research Institute, Sun Yat-sen University, Nanshan, Shenzhen 518087, China

**Keywords:** Shannon entropy, diffusion coefficient, quantum walk

## Abstract

Non-Hermitian topological edge states have many intriguing properties, however, to date, they have mainly been discussed in terms of bulk–boundary correspondence. Here, we propose using a bulk property of diffusion coefficients for probing the topological states and exploring their dynamics. The diffusion coefficient was found to show unique features with the topological phase transitions driven by parity–time (PT)-symmetric non-Hermitian discrete-time quantum walks as well as by Hermitian ones, despite the fact that artificial boundaries are not constructed by an inhomogeneous quantum walk. For a Hermitian system, a turning point and abrupt change appears in the diffusion coefficient when the system is approaching the topological phase transition, while it remains stable in the trivial topological state. For a non-Hermitian system, except for the feature associated with the topological transition, the diffusion coefficient in the PT-symmetric-broken phase demonstrates an abrupt change with a peak structure. In addition, the Shannon entropy of the quantum walk is found to exhibit a direct correlation with the diffusion coefficient. The numerical results presented herein may open up a new avenue for studying the topological state in non-Hermitian quantum walk systems.

## 1. Introduction

Quantum walk [1,2,3,4,5,6] is the quantum analog of the classic random walk, which has found wide applications in many areas of quantum information science. In recent years, quantum walks have been extensively theoretically studied, and their experimental realizations have been reported by single neutral atoms in optical lattices [2], photons in waveguide lattices [7], trapped ions [3,5] and single photons in free space [8]. Since quantum walk can be constructed with various symmetries by designing appropriate evolution operators with periodic driving, it enables the ready observation of the topological phenomena with Floquet methods. Quantum walk has been proven to be suitable for simulating topological materials and exploring dynamics in a wide range [9].

Recently, the use of quantum walks to simulate and explore new topological phenomena [10,11,12] has attracted wide interests. Particularly, a new type of quantum walk, open quantum walks [13,14], have been realized and investigated, where the topological properties can be controlled by dissipation and decoherence. Open quantum walks show rich dynamical behavior [13,14,15,16] and can be used for quantum algorithms in quantum computation for specific tasks [17] and quantum state engineering [13,17]. PT-symmetric quantum walks, as one type of open quantum walks, have recently been investigated both theoretically [18,19] and experimentally [20,21].

However, the physical implementation of PT-symmetric quantum walks have been limited due to a few issues. First is the limited evolution time induced by dissipation and decoherence effects [22]. These effects are unavoidable in open quantum walks [13,14], which is because of the non-unitary dynamics and is described in the framework of the non-Hermitian quantum mechanics [23,24]. Second, PT-symmetric quantum walks and related topological properties have only been studied by the bulk–boundary corres- pondence [25,26,27], while the characterization of bulk property is thus far absent, such as diffusion property and Shannon entropy. Until very recently, the connection between diffusion property and topological property have been studied for skyrmions [28], and the relation between diffusion phenomenon and bulk–edge correspondence has been discussed [29].

In this article, we propose the use of a diffusion coefficient, one of the bulk properties, for characterizing the topological phases in PT-symmetric quantum walk. With this new perspective, we can address the aforementioned two issues and there is no necessity for constructing artificial boundaries. We further illustrate the direct correlation between Shannon entropy and the diffusion coefficient for the first time. The result suggests a fascinating possibility for the exploration of the topological properties of non-Hermitian systems [30,31] using the diffusion coefficient. It is noted that our main tool is numerical simulation instead of the analytical method so that the general principle can be further revealed by analytical calculation.

## 2. Discrete-Time Quantum Walk

The quantum walk is governed by the time-evolution operator [20]:
(1)$$U=\mathrm{LTC}\left(\beta \right)L^{\prime }TC\left(\alpha \right)$$
where *T* is the position operator:
(2)$$T=\sum_{x}\left|x+1\rangle \langle x|⊗|↑\rangle \langle ↑|+|x-1\rangle \langle x|⊗|↓\rangle \langle ↓\right|$$
where *x* is the position of the walker. The position state changes according to the coin states |↑〉 and |↓〉 of the walker. *C* is the position-dependent coin operator:(3)$$C\left(\theta \right)=I⊗\left(\begin{array}{cc} \cos\theta & \sin\theta \\ \sin\theta & -\cos\theta \end{array}\right)$$
where *I* is a unit matrix with the same dimension as the number of grid points. The loss operators are defined as
(4)$$L=I⊗\left[\begin{array}{cc} l_{1} & 0\\ 0 & l_{2} \end{array}\right],L^{\prime }=I⊗\left[\begin{array}{cc} l_{2} & 0\\ 0 & l_{1} \end{array}\right]$$
The evolution operator of the discrete-time quantum walk can be written in the form of $U=e^{-iH_{eff}}$, where $H_{eff}$ represents the equivalent Hamiltonian of the system. Let λ be the eigenvalue of the evolution operator, then we have:(5)$$U|\psi \rangle =\lambda |\psi \rangle ,\lambda =e^{-i\epsilon +\epsilon_{0}}$$
where ϵ is the quasi-energy of the system $e^{-\epsilon_{0}}=1/l_{1}l_{2}$. When $l_{1}=l_{2}=1$, the evolution operator *U* is a unitary matrix with $\epsilon_{0}=0$ and |λ|=1. The system is a Hermitian system. When $l_{1}\neq l_{2}$, it is a non-Hermitian system and *U* is a non-unitary operator, which respects the PT-symmetry $\left(\mathrm{PT}\right)U{\left(\mathrm{PT}\right)}^{-1}=U^{-1}$.

## 3. Diffusion Coefficient and Shannon Entropy

As demonstrated in [20], the topological phase of the evolution operator can be changed by tuning two variables of the system evolution operator α and β. There are boundaries between topological phases with different topological invariants. When $\left(l_{1},l_{2}\right)=\left(1,1\right)$ which corresponds to no loss, the evolution operator is the Hermitian operator. In this case, the topological invariant at the boundary is not defined. When $\left(l_{1},l_{2}\right)=\left(1,0.8\right)$, the evolution operator is a non-Hermitian operator, and there is a boundary area between different topological phases, where the PT-symmetry of the state is broken, and the topological invariant is not defined. It has been proven that regardless of whether a Hermitian operator or PT-symmetric non-Hermitian operator is used to construct an artificial boundary in the system, there will always be a topological edge state at the boundary when evolution operators with different topological invariants are used on each side of the boundary [20], which is related to the bulk–boundary correspondence in the topological effects [32].

In order to study the topological effect of the quantum walk, traditionally, artificial boundaries are constructed by employing different evolution operators on each side of the boundary, then the probability distribution at the boundary is probed, thus enabling the simulation of the bulk–boundary correspondence in topological materials [10]. Recently, the time evolution of the position variance [33,34,35], statistical moments of the walker position distribution [36,37,38], are measured for either investigating the localization effects or revealing the topological phase transitions in discrete-time photonic quantum walks.

Here, we adopt a different approach, that of using the diffusion coefficient [34,39], a natural feature of the quantum walk as the probe of the topological edge states, since the evolution of the system during the quantum walk would lead to a distinctive diffusive behavior [34]. In addition, we investigate the properties of Shannon entropy in the quantum walk. In the following sections, we simulate the quantum walk, both Hermitian and PT-symmetric non-Hermitian, and observe the behavior of the diffusion coefficient and Shannon entropy. In all cases, we take the initial state as |ψ〉=|0〉.

### 3.1. Hermitian Quantum Walk

First, we studied quantum walk with unitary evolution in the Hermitian case. The quasi-energy of the system and the Bloch vector are given by
(6)cosϵ(k)=−sinαsinβ+coskcosαcosβ,
and $\vec{n}\left(k\right)=n_{x}\left(k\right)\hat{i}+n_{y}\left(k\right)\hat{j}+n_{z}\left(k\right)\hat{k}$ with:
(7)$$\begin{array}{c} n_{x}\left(k\right)=\frac{\sin k\sin\alpha \cos\beta }{\sin\epsilon (k)}\\ n_{y}\left(k\right)=\frac{\cos\alpha \sin\beta +\cos k\sin\alpha \cos\beta }{\sin\epsilon (k)}\\ n_{z}\left(k\right)=\frac{-\sin k\cos\alpha \cos\beta }{\sin\epsilon (k)} \end{array}$$
Thus, we obtain the winding number with $\Gamma =\left(\cos\alpha ,0,\sin\alpha \right)$ according to:
(8)$$W=\frac{1}{2\pi }\oint dk\left(\vec{n}\times \frac{\partial \vec{n}}{\partial k}\right)\cdot \Gamma$$
The resulting phase diagram for the winding numbers is shown in Figure 1a. We observed two topological phases corresponding to W=0 and W=1 which are separated by phase-transition lines. Red and blue lines represent the closing of the quasienergy gap at ϵ=0 and ϵ=π, respectively. We simulate the topology of the quantum walk on a one-dimensional lattice. Let the initial state of the walker be |ψ〉=|0〉⊗|↑〉, i.e., the initial position of the walker is x=0, and initial coil state is |↑〉.

Then, the calculation of the diffusion coefficient is carried out with a different number of steps *t*. The diffusion coefficient *D* is calculated according to [34,39]:
(9)$$D=\frac{\sigma^{2}\left(t\right)}{2t}=\frac{M_{2}\left(t\right)-M_{1}^{2}\left(t\right)}{2t}$$
where $M_{1}\left(t\right)=\sum_{n=-t}^{n=t}nP_{n}\left(t\right)$ and $M_{2}\left(t\right)=\sum_{n=-t}^{n=t}n^{2}P_{n}\left(t\right)$ are the first and second moment, $P_{n}\left(t\right)$ is the probability that the position of the walker is n at time t. $\sigma^{2}$ is the variance of the quantum walk. Figure 1b shows the results of t=5, t=15 and t=50 with the same β=π/4. For all three chosen t, they show a very similar trend with α, although the diffusion coefficient increases with t. In the topological phase of W=0, the diffusion coefficient is quite stable with small fluctuations, while in the topological phase of W=1, it shows a valley structure along with α. More importantly, in the close proximity of the topological phase transition, a turning point appears on the diffusion coefficient with the change of α. In Figure 1c, the dependence of the diffusion coefficient on the spin–rotation angle θ1 is given for β=π/6, β=π/4 and β=π/3 with the number of steps t=15, respectively. On the bottom α−β plane, phase-transition lines are shown as Figure 1a, together with a vertical dashed line in order to clearly distinguish those turning points in the vicinity to the topological phase transition. It was also noticed that the diffusion coefficient decreases with the increase in β for the same α.

Subsequently, we obtain the corresponding Shannon entropy of the walker according to:
(10)$$S=-\sum_{i=1}^{m}P_{n}\left(x_{i}\right)\log_{2}P_{n}\left(x_{i}\right)$$
where $P_{n}\left(x_{i}\right)$ is the probability that a quantum walker is in position $x_{i}$ at time *n*. The results are illustrated in Figure 1d. It is clear and interesting to determine that the Shannon entropy amazingly resembles the diffusion coefficient in terms of the dependence on spin–rotation angles α and β, probably due to both of them being related to $P_{n}\left(t\right)$.

### 3.2. PT-Symmetric Non-Hermitian Quantum Walk

In particular, we are more interested in the behavior of a non-Hermitian quantum walk, where the quantum coin (or position) carried by the walker interacts with an environment, resulting in a loss. Such a system would allow us to explore topological properties of non-Hermitian systems and to investigate Floquet topological phases [18,20]. Moreover, it is shown that the quantum walk properties are highly sensitive to decoherent events. As the quantum coil (or position) is coupled to an environment, decoherence would lead a transition from quantum walk to classical random walk for a large number of steps, and a relation between the decoherence and the diffusion properties of the system can be established [33,34,40,41]. In addition, a weak decoherence could possibly enhance the properties of the quantum walk and is thus beneficial to the development of quantum algorithms [41].

Here, we consider the PT-symmetric non-Hermitian scenario by calculating $\left(l_{1},l_{2}\right)=\left(1,0.8\right)$ with the same calculation and analysis. We calculate the diffusion coefficients for a different number of steps for t=5, t=15 and t=30 three different number of steps with the same β=π/4, the results are shown in Figure 2a. Based on the phase diagram of the PT-symmetric non-Hermitian quantum walk [20], the diffusion coefficient is found to show similar features to Figure 1b, which is exhibiting an abrupt change at the boundary of the topological phase transition. As α increases from −π to 0, the system starts from PT-symmetric phase (corresponding to a topologically non-trivial state with the topological number $\left(\nu_{0},\nu_{\pi }=\left(1,1\right)\right)$, and transits broken PT-symmetric phase which is represented by the red region in the vicinity of α=−3π/4, with which the real part of the eigenenergy is π. Then, the system enters another PT-symmetric phase (a topologically non-trivial state with topological number $\left(\nu_{0},\nu_{\pi }=\left(1,-1\right)\right)$, and transits previous broken PT-symmetric phase again, finally reaches a PT-symmetric phase (a topologically non-trivial state with topological number $\left(\nu_{0},\nu_{\pi }=\left(1,1\right)\right)$. The diffusion coefficient stays stable until it reaches the broken PT-symmetric phase. When it first crosses the broken PT-symmetric phase, the diffusion coefficient gradually increases before α=−3π/4 and then decreases, exhibiting a peak-valley profile. When it crosses the broken PT-symmetric phase again, the diffusion coefficient increases before α=−π/4 and then decreases, exhibiting a valley–peak profile. Variations on the diffusion coefficient from α=0 to π follow the same path as from α=−π to 0.

Another feature of the diffusion coefficient is that its amplitude is much smaller than that in the Hermitian case, and it decreases with the increase in the number of the step t due to the existence of the loss operator. Figure 2b illustrates the dependence of the diffusion coefficient on the spin–rotation angle α for β=π/6, β=π/4 and β=π/3 with the number of steps t=15, respectively. We chose t=15 because of several reasons. First, we find that the effects of varying t are much larger than those obtained by varying α. Second, a larger *t* would lead to a too small value of the diffusion coefficient to observe any effect due to excessive loss. Third, the effect of coherent diffusion would be invisible if the number of steps t was very small. Therefore, in order to better observe the effect of PT-symmetric quantum walk with the topological phase in the non-Hermitian scenario, we take t=15 for the following calculations for both the diffusion coefficient and Shannon entropy. From Figure 2a,b, we notice that the peak profile of the diffusion coefficient has direct correspondence with the broken PT-symmetric phase, i.e., the topological edge state, which suggests a possible intrinsic relation between them. In Figure 2c, Shannon entropy as the function of the spin–rotation angle α for β=π/6, β=π/4 and β=π/3 with a number of steps t=15 are presented, respectively. It is obvious to find that all of them exhibit an interesting resemblance to that for the diffusion coefficient.

## 4. Conclusions

We proposed using the diffusion coefficient as a probe for studying the dynamics of topological edge states. We conducted simulations of both Hermitian quantum walk and PT-symmetric non-Hermitian quantum walk, and investigated the topological phases driven by such quantum walks. The properties of the diffusion coefficient as a function of two parameters α and β in the evolution operator of the quantum walk are studied. It has been found that the diffusion coefficient exhibits qualitatively similar characteristics as the topological phase transition, which appears in Hermitian and PT-symmetric non-Hermitian quantum walks. For a Hermitian system, a turning point and abrupt change appear in the diffusion coefficient when the system is at the boundary of the topological phase transition, while it remains stable in the trivial topological state. For a PT-symmetric non-Hermitian system, however, the diffusion coefficient displays unique peak structure in the topological phase transition region. In addition, Shannon entropy is found to illustrate very corresponding behavior such as the diffusion coefficient. Such interesting signatures might give us a unique probe for the topological edge states, and suggest exciting possibilities for studying the topological boundary effects and edge states in PT-symmetric non-Hermitian quantum walks by exploiting diffusion coefficient and Shannon entropy.

Moreover, we envision one experiment that could be performed in the near future, which drives a single-trapped ${}^{171}{\mathrm{Yb}}^{+}$ ion with a pair of Raman beams to control the ions momentum state and uses the ions momentum state and internal hyper-fine energy levels in ground electronic state as the position and coin state, respectively. A dissipation laser which drives an electronic transition can induce particle loss in the hyperfine ion qubit and create a parity–time symmetric open system. With this configuration, the diffusion coefficient and the Shannon entropy simulated in this work can be experimentally investigated.

## Figures and Tables

**Figure 1 entropy-23-01145-f001:**
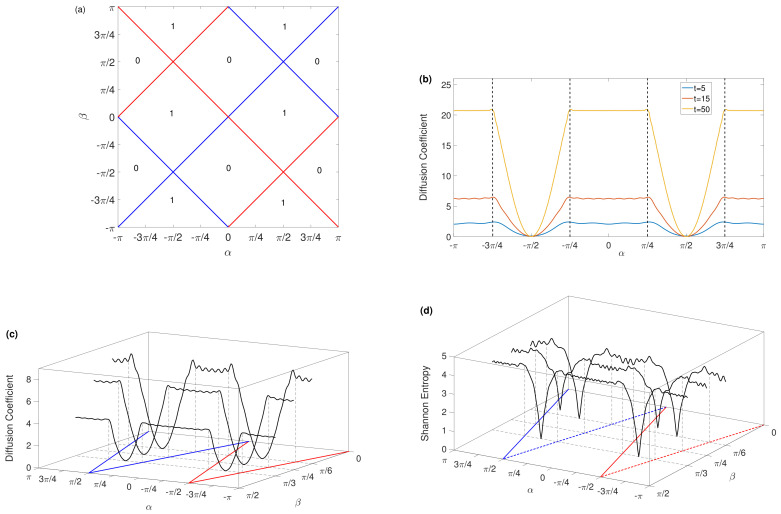
(**a**) Phase diagram for discrete time quantum walk in unitary evolution, where winding numbers are a function of the spin–rotation angles *α* and *β*. Topologically distinct gapped phases characterized by the winding number W are separated by phase-transition lines where the quasienergy gap closes at either *E* = 0 (red) or *E* = *π* (blue). (**b**) The diffusion coefficient of the quantum walk in unitary evolution with different number of steps for the same *β* = *π*/4. Blue, red and yellow lines correspond to *t* = 5, *t* = 15 and *t* = 50 steps, respectively. (**c**) The dependence of the diffusion coefficient on the spin–rotation angles *α* and *β* with number of steps *t* = 15. (**d**) The dependence of Shannon entropy on the spin–rotation angles *α* and *β* with number of steps *t* = 15.

**Figure 2 entropy-23-01145-f002:**
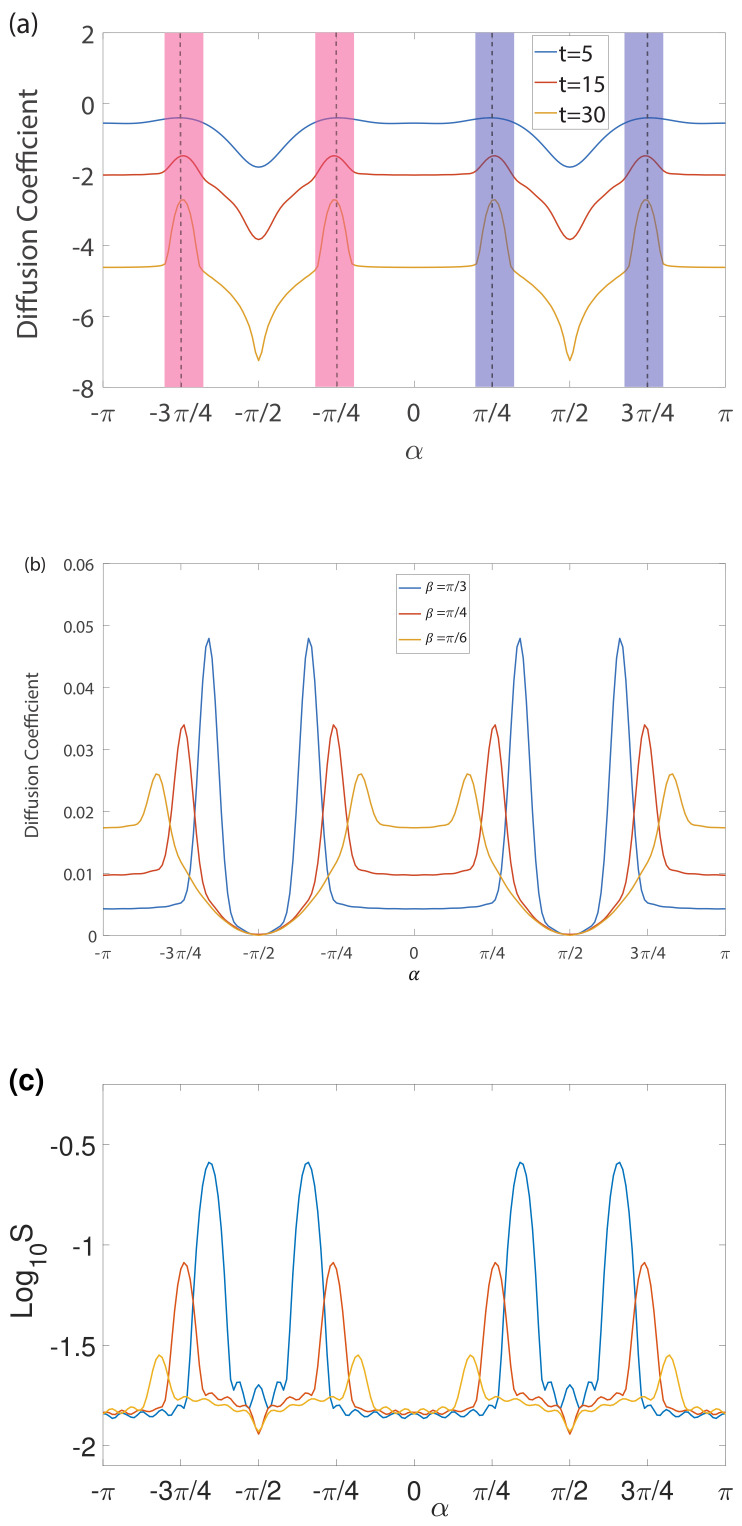
(**a**) The diffusion coefficient of the quantum walk in the PT-symmetric non-unitary evolution with a different number of steps for the same *β* = *π*/4. Blue, red and yellow lines correspond to *t* = 5, *t* = 15 and *t* = 30 steps, respectively. The blue and red shaded regions represent broken PT-symmetric phases [20] with complex eigenenergies whose real parts are 0 and *π*, respectively. (**b**) The dependence of diffusion coefficient on the spin–rotation angle *α* with number of steps *t* = 15 for *β* = *π*/6, *β* = *π*/4 and *β* = *π*/3, respectively. (**c**) The dependence of Shannon entropy on the spin–rotation angle *α* with number of steps *t* = 15 for *β* = *π*/6, *β* = *π*/4 and *β* = *π*/3, respectively.

## Data Availability

Not applicable.

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
