# Peer review of "Shannon Entropy and Diffusion Coefficient in Parity-Time Symmetric Quantum Walks"

_entropy, 2021, doi:10.3390/e23091145_

Round 1

Reviewer 1 Report

This manuscript investigates by numerical simulations a coined quantum walk and tries to link the diffusion (measured by the walker's second moment) with an angle parameter related to the topology of the system in the band structure of the underlying effective Hamiltonian. In the following, I list the many flaws of this manuscript, which I must reject as a natural consequence:

1) "quantum random walks" I read in the introduction. Quantum walks are fully deterministic (as long as no precise measurements are discussed). Hence, no randomness at all!

2) In the 2nd paragraph of the Intro, the problem of exponentially growing resources must be mentioned when you want to use a walk as a universal quantum computer. Hence, practically, the system might be judged as useless.

3) In the entire introduction, I am reading nice and fashionable bus words, but little concentration and focus on what is actually studied here. What has been done before in the direction of THIS study? The introduction must be completely rewritten, putting into context the presented study and results, and saying explicitly what is new here and has never been touched in other works.

4) The angles theta_1,2 are different from the coin angle theta in eq. (3). The notation is confusing here. use another symbol for one fo the two angles.

5) The central result of this work seems to me fig. 2. Now this figure is much too small, labels are unreadable. Please improve and rescale to at least the double size.

6) The results are purely numerical without any analytical insight that might be used for predictions on many setups without redoing the simulations for each case.

7) No experimental implementation is discussed. How difficult would it be to actually realise and see similar results as in fig. 2? This point goes in full line with issue 3) above.

8) Following issue 7), a key reference on an experimental realisation is missing that seems nearest to th where investigated Floquet engineered walks: Phys. Rev. Lett. 121, 070402 (2018).

Author Response

We appreciate the referee pointing out “This manuscript investigates by numerical simulations a coined quantum walk and tries to link the diffusion (measured by the walker's second moment) with an angle parameter related to the topology of the system in the band structure of the underlying effective Hamiltonian.” 

The referee list flaws of this manuscript, and intend to reject. However, in current version, we correct all the flaws as following. So, as a natural consequence, we request the first referee to reconsider.

The comments are:

1)"quantum random walks" I read in the introduction. Quantum walks are fully deterministic (as long as no precise measurements are discussed). Hence, no randomness at all!

We agree with the referee that in regards of the value of the probability it is deterministic if no measurement is performed, otherwise the probability would collapse which shows the randomness. We replace all the “quantum random walks” with “quantum walk”.

2)In the 2nd paragraph of the Intro, the problem of exponentially growing resources must be mentioned when you want to use a walk as a universal quantum computer. Hence, practically, the system might be judged as useless.

We delete all the over claim words about quantum walk as a universal quantum computer. We agree that employ the quantum walk as the quantum algorithm for performing universal quantum computation need exponentially growing resources. However, for some specific computational tasks, quantum walks with a limited number of walks can be employed with finite computational resources. For example, by employing single trapped ion for performing quantum walk and using the ion’s momentum state and internal hyperfine energy levels. So, we only emphasize in the second paragraph of the introduction part “…can be used for quantum algorithms in quantum computation for specific tasks [21]”.

3)In the entire introduction, I am reading nice and fashionable bus words, but little concentration and focus on what is actually studied here. What has been done before in the direction of THIS study? The introduction must be completely rewritten, putting into context the presented study and results, and saying explicitly what is new here and has never been touched in other works.

Following your suggestions, we have completely rewritten the introduction. In the 4th paragraph of the introduction we specify what we have done in this paper “In this article, we propose to use diffusion coefficient, one of the bulk properties, for characterizing the topological phases in PT-symmetric quantum walk. With this new viewpoint, we can address the above-mentioned two issues and there is no necessity of constructing artificial boundaries. We further illustrate the direct correlation between Shannon entropy and diffusion coefficient for the first time. The result suggests a fascinating possibility for exploration topological properties of non-Hermitian systems using diffusion coefficient. It is noted that our main tool is the numerical simulation instead of the analysis method so that the general principle can be further revealed by analysis calculation.”

4)The angles theta_1,2 are different from the coin angle theta in eq. (3). The notation is confusing here. use another symbol for one of the two angles.

The angles and in Equation (1) are different coin angles just as the coin angle in Equation (3). In order to better distinguish between them, we replace and with and in the main context.

5)The central result of this work seems to me fig. 2. Now this figure is much too small, labels are unreadable. Please improve and rescale to at least the double size.

We have resized all three graphs in Figure 2 as suggested by the referee.

6)The results are purely numerical without any analytical insight that might be used for predictions on many setups without redoing the simulations for each case.

We admit that this letter is a simulation mainly on non-Hermition quantum walk and is not motivated to provide analytical results. We hope further investigation could reveal general analytical principle corresponding to our numeric results. We emphasize this point in the paper “It is noted that our main tool is the numerical simulation instead of the analytical method so that the general principle can be further revealed by analytical calculation.”

7)No experimental implementation is discussed. How difficult would it be to actually realise and see similar results as in fig. 2? This point goes in full line with issue 3) above.

In the conclusion part, we have envisioned one experiment could be done in our group in the near future, which drives single trapped 171Yb+ ion with a pair of Raman beams to control the ion’s momentum state and uses the ion’s momentum state and internal hyperfine energy levels in ground electronic state as the position and coin state, respectively. A dissipation laser which drives an electronic transition can induce the particle loss in the hyperfine ion qubit and create a Parity-Time symmetric open system.

8) Following issue 7), a key reference on an experimental realisation is missing that seems nearest to th where investigated Floquet engineered walks: Phys. Rev. Lett. 121, 070402 (2018).

We have added the mentioned paper in the references.

Reviewer 2 Report

The notion of quantum walks is very useful to implement quantum algorithms and finds also immediate applications in modeling a diversity of phenomena in quantum optics and quantum communications. Theoretical approaches show that quantum walks facilitate the exploration of emerging topologies, mainly in non-Hermitian models where the parity-time symmetry holds. In the present work the diffusion coefficient is suggested as a convenient parameter to characterize topological phases without requiring concrete boundaries. Quite interestingly, such a coefficient is shown to be in direct connection with the Shannon entropy. The work can be extended to explore the topological boundary effects in quantum walks for systems where the PT-symmetry is broken.

I find the paper to be well written, and includes results that may be interesting for the Entropy readers. I recommend accepting it for publication as is.

Author Response

We appreciate the referee’s opinion “I find the paper to be well written, and includes results that may be interesting for the Entropy readers. I recommend accepting it for publication as is.”

Reviewer 3 Report

I have carefully read the manuscript entropy-1274499. The Authors study dynamics of topological edge states by analyzing the
corresponding diffusion coefficient and the Shannon Entropy. They
consider both Hermitian quantum walk and PT-symmetric non-Hermitian
quantum walk. They have proved that the diffusion coefficient exhibits
qualitatively similar characteristics at the topological phase
transition both for the case of Hermitian and PT-symmetric non-Hermitian
quantum walks. For a Hermitian system, they have shown that a turning
point and abrupt change appears in the diffusion coefficient when the
system is at the boundary of the topological phase transition, while it
remains stable in trivial topological state. For a PT-symmetric
non-Hermitian system, however, the diffusion coefficient displays a
unique peak structure in the topological phase transition region. In
addition, they have found that Shannon entropy is in correspondence
with the behaviour as the diffusion coefficient. The work is clearly presented, and I think that the results are of
great interest in the area of topological systems.
I have a few comments: 1- In the non-hermitian case, the eigenstates of $H_{eff}$ are
non longer orthogonal. Also, $U$ is non longer unitary. Concerning
Eqs. (9) and (10), in the non-hermitian case, what is the explicit
expression used to compute the elements of the density matrix?. 2- Concerning the statement of line 67 on page 3, to my knowledge,
PT-symmetry breaking is related to the break of the symmetry of the
states not of the operators (the operator doesn't change, the states
behave differently in the different phases). 3- Concerning the references, perhaps the Authors would like to
include a recent work, Sci. Reports 11, 10262 (2021). In my opinion, the manuscript provides an original contribution to
the study of non-hermitian quantum walks. However, before I can
recommend the manuscript for publication in Entropy, the previous
issues should be addressed.  

Author Response

We appreciate the referee’s opinion “The work is clearly presented, and I think that the results are of great interest in the area of topological systems.” 

We have corrected the paper to address all these comments.

1) In the non-hermitian case, the eigenstates of $H_{eff}$ are non longer orthogonal. Also, $U$ is non longer unitary. Concerning Eqs. (9) and (10), in the non-hermitian case, what is the explicit expression used to compute the elements of the density matrix?

In the non-Hermitian case, the simulation was done in the same way with the Hermitian case. The diffusion coefficient D and the Shannon entropy S were calculated according to the probability, which was calculated using the corresponding non-Hermitian operator expressed in Equation (5) with l_1=1 and l_2=0.8 instead of compute the density matrix. We emphasize the calculation tool is numerical simulation instead of finding density matrix by the last sentence in the 1st paragraph of the introduction, “It is noted that our main tool is the numerical simulation instead of the analytical method so that the general principle can be further revealed by analytical calculation.”

2) Concerning the statement of line 67 on page 3, to my knowledge, PT-symmetry breaking is related to the break of the symmetry of the states not of the operators (the operator doesn't change, the states behave differently in the different phases).

The referee’s comment is absolutely right about “PT-symmetry breaking is related to the break of the symmetry of the states not of the operators”. We made a mistake in the statement in the last version and we have corrected this statement in the main context.

3) Concerning the references, perhaps the Authors would like to include a recent work, Sci. Reports 11, 10262 (2021). In my opinion, the manuscript provides an original contribution to the study of non-hermitian quantum walks. However, before I can recommend the manuscript for publication in Entropy, the previous issues should be addressed.

We have added the mentioned paper in the references.

Round 2

Reviewer 3 Report

In my opinion, the manuscript provides an original contribution to the understanding of quantum walks and their applications. The Authors have addressed the issues I have previously quoted.

I recommend the manuscript in Entropy.